# Topical Formulation of Nano Spray-Dried Levocetirizine Dihydrochloride against Allergic Edema

**DOI:** 10.3390/pharmaceutics14122577

**Published:** 2022-11-23

**Authors:** Mirella Mirankó, Judit Tóth, Andrea Fodor-Kardos, Krisztina Móricz, Antal Balázs Szenes-Nagy, Attila Gácsi, Tamás Spaits, János Gyenis, Tivadar Feczkó

**Affiliations:** 1Institute of Materials and Environmental Chemistry, Research Centre for Natural Sciences, Magyar Tudósok Körútja 2, 1117 Budapest, Hungary; 2Faculty of Engineering, Research Institute of Biomolecular and Chemical Engineering, University of Pannonia, Egyetem u. 10, 8200 Veszprém, Hungary; 3EGIS Pharmaceuticals PLC, Keresztúri út 30-38, 1106 Budapest, Hungary

**Keywords:** levocetirizine dihydrochloride, nano spray drying, topical oleogel, allergic edema

## Abstract

Levocetirizine dihydrochloride active ingredient was microencapsulated using nano spray-drying technology for preparing microparticles containing topical gel against edema. Hydroxyl propyl methyl cellulose (HPMC) was used as a carrier polymer during spray drying. The active ingredient content of the nano spray-dried products was 52.81% (*w*/*w*) and 51.33% (*w*/*w*) for ex vivo and in vivo experiments, respectively, and the average particle size was 2.6 µm. X-ray diffraction analysis indicated an amorphous state of the active ingredient embedded in the amorphous matrix of the polymer. Dermal oil gels composed of Miglyol 812 gelated by Dermofeel viscolid included 5% (*w*/*w*) (for ex vivo) and 10% (*w*/*w*) (for in vivo) active ingredient without or with 0.05% (*w*/*w*) menthol penetration enhancer. Qualitative ex vivo penetration studies using a confocal Raman microscopic correlation mapping were executed on human abdominal skin. The results showed that the active ingredient was enriched in the epidermis and upper dermis layer of the skin using oleogel loaded with the nano spray-dried drug-HPMC composite. Menthol addition to the oleogel resulted in the concentration of levocetirizine in the dermis. In vivo tests were performed on a mouse model of croton oil-induced ear edema. Negative control and Fenistil-treated groups were compared using the prepared oil gels with and without menthol. Without penetration enhancer, 20 µL of our oil gel loaded with nano spray-dried levocetirizine dihydrochloride composite showed similar effectiveness to the same volume of Fenistil gel, while 5 µL menthol containing sample was sufficient to eliminate the skin irritation similarly to 20 µL Fenistil.

## 1. Introduction

Angioedema is an area of swelling (edema) of the lower layer of skin and tissue just under the skin or mucous membranes [1]. In the case of urticaria, edema and vasodilatation in the upper and middle part of the dermis can be seen histologically, with perivascular inflammatory cell infiltration and sporadic mast cell proliferation [2]. Histamine-related angioedema and urticaria can be treated with antihistamines and corticosteroids [1]. Cetirizine is a second-generation antihistamine that reduces the natural chemical histamine in the body [3]. It is used to temporarily relieve the symptoms of hay fever and allergy to other substances (such as dust mites, animal dander, cockroaches, and molds). Cetirizine chemically contains a chiral center and commercially is a racemic mixture of levocetirizine (R-(−) cetirizine) and dextrocetirizine (S-(+) cetirizine) [4]. The R-enantiomer of racemic cetirizine is the pharmacologically active enantiomer, which has higher affinity for human H1 receptors than its racemic mixture [5]. Oral antihistamines are commonly used to decrease allergic symptoms [6], although they can cause side effects as drowsiness and fatigue [7]. These symptoms can be avoided by topical medications, which also work best for skin conditions and quick treatment for joint pain. Gels are defined as three-dimensional polymer networks swollen by large amounts of solvent and have been classically used as topical drug-delivery systems [8]. Gels are classified based on the nature of solvent as hydrogel, xerogel, and organogel (oleogel) [9]. Oleogels are promising drug formulations concerning their physical and chemical stability and increased in vivo transdermal efficacy [10,11]. They ensure simplicity in preparation compared to hydrogels. Some studies investigated different types of semisolid formulations as antihistamines using cetirizine as active ingredient. H. Walch in patent application [12] dealt with the topical application of cetirizine and loratidine and their salts, preparing different hydrophilic and hydrophobic (greasing) gels. They used different pharmaceutical excipients and solutions to enhance the weak skin penetration of the active ingredients. Ciurlizza et al. [13] prepared hydrogel and nanoemulsion for the topical delivery of cetirizine. The samples were tested for human skin permeation by Franz diffusion cell and antihistaminic activity on rabbits, and both of the formulations showed effectiveness comparable to the two commercially available products Fenistil^®^ and Polaramine^®^. Subramanian et al. [14] prepared transdermal patches by the solvent evaporation method containing cetirizine with hydroxyl propyl methyl cellulose, polyvinyl pyrolidine, and ethyl cellulose polymer carriers. There was no report about the skin permeation abilities of the formulations. Majumber et al. [15] produced chemically modified amine salts of cetirizine as a supramolecular gel. Gels were formed by methylsalicilate and with or without 1% of menthol. The methylsalicylate/menthol topical gel produced from salt with tyramine showed excellent in vivo self-delivery applications in treating dinitrochlorobenzene-induced allergic ear redness and skin-contact hypersensitivity in mice. Although topical drug therapy is associated with a benefit of reducing the risk of systemic side effects, the drug can induce local irritation [16]. Particle technology is one of the solutions to improve treatment tolerability and to use a lower dosage of drugs [17]. Microspheres are biologically inert polymer particles that bind drugs and release the active molecules in certain time. Spray drying is one of the processes for preparing pure and composite microparticles [18,19]. Amelian et al. [20] prepared methacrylate-based microparticles of cetirizine dihydrochloride by spray drying for taste masking. The obtained particle size was between 4.7 and 8.6 µm dependent on the concentration of the solutions and the polymer/drug ratio.

It can be established based on the literature data dealing with cetirizine topical application that in previous studies the LC was in a dissolved form in different types of hydrogel or oleogels with substantial water content. In those cases, the formulation requires several excipients, such as surfactant, preservative, co-solvent, and penetration enhancer, to reach the expected skin penetration. However, in our study we developed oleogels without adding water and/or surfactants. The commercially available products of cetirizine are solid, drop, or syrup formulas. For example, ZYRTEC^®^ Liquid Gels is a capsule of 10 mg/capsule content for oral use [21]. Our aim was to prepare a new gel formulation of levocetirizine dihydrochloride for topical administration against allergic edema-like urticaria. Oleogel was chosen as the base for the product as a promising vehicle for transdermal delivery. Composite microparticles with HPMC polymer were prepared by a nano spray dryer to ensure the easy incorporation of the solid-state drug in oleogel. Gels were prepared without and with menthol as a penetration enhancer for dermal administration. Menthol is a common excipient in different cosmetics and topical drug formulations. It has a dual effect: it acts as a local anesthetic and a penetration enhancer [22,23]. The ex vivo cutaneous penetration of drugs was investigated on human skin samples by confocal Raman microscopic method, and in vivo tests were performed on a mouse model of croton oil-induced ear edema. The efficacy was compared to commercially available Fenistil gel containing dimetindene maleate as the active ingredient.

## 2. Materials and Methods

### 2.1. Materials

Levocetirizine dihydrohloride (LC), Miglyol 812 and Dermofeel viscolid were a kind gift from Egis Pharmaceuticals PLC (Budapest, Hungary). Methocel E5 (Hydroxypropyl methylcellulose, HPMC) were purchased from Colorcon Ltd. Menthol (racemic) was delivered by Alfa Aesar, Hungary. Acetone was obtained from Reanal Ltd. (Budapest, Hungary) and croton oil was obtained from Sigma-Aldrich. Fensitil gel is marketed by GSK (Brentford, UK). Cyclohexane (purum) is a product of Molar Chemicals Ltd. (Halásztelek, Hungary). Soy lecithin was food-grade.

### 2.2. Nano Spray-Drying Experiments

The HPMC-drug composite microparticles were prepared by Nano Spray Dryer B-90 (Büchi Labortechnik AG, Flawil, Switzerland). The operation of the spray dryer can be reached elsewhere in detail [24]. The main principle is the following: the drying is carried out with preheated gas entering on the top of the chamber. Droplet generation is based on microatomization technology, i.e., a piezoelectric actuator driven by an electronic circuit vibrates the thin stainless-steel mesh in the exchangeable spray cup. The feeding of the solution is regulated by the recirculation pump and spray rate. The electrostatic collector collects the solidified droplets as fine particles from the leaving gas. The experiment conditions were the following: the drying gas flow rate was 90 L/min with an inlet temperature of 100 °C. A membrane with a 7 µm mesh was used for droplet generation. The recirculation pump rate and the spray rate were 60% and 35%, respectively. The drying experiment was carried out using the tall set-up employed for water-based samples. Solutions for the experiments were prepared as follows: 1% (*w*/*w*) of LC and 1% (*w*/*w*) of HPMC and the MilliQ water were weighed on analytical scale and the solutions were prepared by magnetic stirring and used without filtering.

### 2.3. Characterization of Spray-Dried Samples

The active ingredient content in the dried samples was investigated by spectrophotometric method using Shimadzu UV-1800 instrument (Kyoto, Japan). To determine the drug content in the dried samples, calibration curve was recorded for the concentrations 5, 10, 20, 30, and 40 μg/mL. The equation of the calibration curve is as follows (Equation (1)):(1)$$C_{LC}=\frac{\left(A_{230.5}+0.0011\right)/0.0321}{40}\times 100,$$
where *C_LC_*: *LC* content % (*w*/*w*) in the dried samples.

The particle size and distribution were measured by laser diffraction method. The optical parameters for the calculation (provided by the Mastersizer 2000 software update v6.01) were the following: refractive index 1.52 and absorption coefficient 0.5. Using these data, the residual value was 1% according to the Malvern Instruments Operators Guide [25]). The particle size was reported as a volume equivalent sphere diameter marked as D(4,3), and the distribution by Dv90, Dv10, and Dv50 data are the 90%, 10%, and 50% cumulative volume distributions, respectively. The particle size distribution was also characterized by span data (Dv90-Dv10)/Dv50), which are used to calculate the size distribution width. The measurements were carried out by a Malvern Mastersizer 2000 instrument (Malvern Instruments, Malvern, UK) using the SM dispersion unit with a stirring rate of 2000 rpm. A total of 10 mg dried sample was added into 1 mL of cyclohexane solution containing 0.1% (*w*/*w*) soy lecithin and sonicated for 40 s at 30% of power with 6 mm probe by Sonics VCX 130 sonicator, then the obtained suspension was filled into the dispersion unit for measurement. The particle size of the microparticles was also measured in the prepared gels: the gel was dissolved in cyclohexane containing 0.1% (*w*/*w*) soy lecithin to investigate whether its size changed during the incorporation in gel.

X-ray diffraction images were recorded using a Philips PW 3710 diffractometer (Philips Analytical, Almelo, The Netherlands) with CuKα radiation with a tube current of 40 mA and a voltage of 50 kV at a scanning rate of 0.02° 2θ/s. Control of the device and data collection was performed with Philips X’Pert Data Collector software.

The morphology of the samples was investigated with a FEI Thermofisher Apreo S (Thermo-Fisher Scientific, Waltham, MA, USA) scanning electron microscope. The spray-dried powder was coated on the grid without additional treatment and 5 kV accelerating voltage was used.

### 2.4. Preparation of Oleogels

For the ex vivo experiments, 10% (*w*/*w*) Dermofeel viscolid powder, 90% (*w*/*w*) Miglyol 812 and SD1 dried sample were used with an active ingredient content of 52.81% (*w*/*w*). In the gel samples the final drug concentration was adjusted to 5% (*w*/*w*). Miglyol 812 and Dermofeel viscolid powder were mixed in a beaker in a 50 °C water bath. The measured amount of SD1 was added into the gel and intensively stirred (1100 rpm) with magnetic stirring for 2 min. Finally, the water bath was removed and the sample was stirred (500 rpm) for 20 min at room temperature. During this time, the sample solidified (LO5) and a homogeneously distributed oleogel was formed. In the case of samples containing penetration enhancers, (LO5M), 0.05% (*w*/*w*) menthol was mixed into the sample after the water bath was removed, before the gel solidification. Samples for in vivo tests were prepared using SD2 sample with a drug content of 51.33% (*w*/*w*). The drug content was adjusted to 10% (*w*/*w*) in the oleogel for in vivo test (LO10). The composition of LO10M was similar; additionally, it contained 0.05%(*w*/*w*) menthol. For the in vivo experiments, 15% (*w*/*w*) Dermofeel viscolid powder and 85% (*w*/*w*) Miglyol 812 were used, because due to the higher drug content, a higher ratio of Dermofeel viscolid was needed to obtain a homogenous oleogel. The samples were stored at 0–5 °C.

### 2.5. Ex Vivo Raman Microscopic Investigations

Raman spectroscopy has been used for skin analysis for 20 years as a well-established technology for investigating the skin with high spatial resolution [26]. Bakonyi et al. [27] investigated the penetration profiles of four different lidocaine-containing formulations (hydrogel, oleogel, lyotropic liquid crystal, and nanostructured lipid carrier) by Raman microscopic mapping of the drug. They concluded that the Raman spectroscopy as a nondestructive technique is an applicable tool for investigating skin distribution and tracking penetration pathways of active agents.

The ex vivo skin penetration investigations were made on the human abdominal skin of a 42-year-old female. The same skin sample was used for each treatment. A total of 100 mg of the formulation was uniformly measured on 1 × 1 cm area of the skin samples. The skin samples were placed in a Petri dish containing cotton wool and filter paper soaked in physiological salt solution, which prevents the skin sample from drying out. The skin samples were stored at room temperature, and the treatment time was 3 h. After the treatment time, the formulations were gently removed from the skin surface, and then the treated surface was cut out with a scalpel. Microscopic sections were prepared from the excised skin samples, for which a Leica CM1950 type cryomicrotome was used. During sectioning, the internal temperature of the sectioning head and the cryomicrotome was set to −23 °C; the thickness of the sections was 15 µm in all cases. The prepared sections were placed on a microscope slide with a matte aluminum foil.

A Thermo Scientific DXR Raman confocal microscope was used to record the Raman spectra of the active substances and formulations. The wavelength of the laser source used was 780 nm, and the laser power for the measurements was 24 mW. The laser light source was guided onto the sample through a 25 µm slit aperture and a 10× magnification objective. A spectrum was recorded for 5 s, and the final spectrum was obtained by averaging a total of 24 recorded spectra.

During the mapping, the individual spectra were recorded using the same instrument and method as used for recording the spectra of active substances and formulations. In the case of skin samples, a 100 × 500 µm area was examined when recording maps, with 50–50 µm steps in the x and y directions. Thus, a total of 44 spectra were recorded to create a correlation map; the recording time of the entire mapping was an hour and a half. During the evaluation, an automatic baseline correction was performed on the recorded spectra.

### 2.6. Animals

Male NMRI mice (25–35 g) were obtained from Toxi-Coop Ltd. (Budapest, Hungary) and were maintained under temperature-controlled conditions (22 +/− 2 °C) with a normal 12 h light–dark cycle and ad libitum access to conventional laboratory diet and tap water. All mouse study protocols were approved by the Animal Care and Use Ethical Committee of Egis Pharmaceuticals PLC and in accordance with the Hungarian Law of Animal Care and Use (1998. XVIII) and Directive 2010/63/EU on the protection of animals used for scientific purposes.

### 2.7. In Vivo Tests

The anti-inflammatory effect of LO10 and LO10M topical formulations was evaluated in croton oil-induced edema in mice. Male NMRI mice (*n* = 8/group) were treated with the formulations in different amounts (5, 10, or 20 µL) on the inner side of the right ear. As positive control, Fenistil gel (20 µL, GSK Consumer Kft.) treatment was used, while as negative control, the treatment was imitated with forceps. After half an hour, edema was induced on the right ear with local application of croton oil in acetone (10 µL 1% solution), and the left ear received vehicle treatment. The thickness of each ear was measured after four hours using a digital caliper (Mitutoyo Corporation, Kawasaki, Japan). The degree of edema was determined by the difference of thickness between the croton oil-treated right ear and the vehicle-treated left ear (mean ± SEM is shown). Statistical evaluation was performed comparing the negative control group with the positive control group using unpaired *t*-test (# *p* ≤ 0.05) and comparing the negative control group with the treated groups using one-way ANOVA followed by Dunnett’s multiple comparison test (* *p* ≤ 0.05).

## 3. Results and Discussion

### 3.1. Characteristics of the Spray-Dried Samples

The average particle size of the dried products was 2.67 and 2.59 µm of SD1 and SD2, respectively (particle size distribution data summarized in Table 1). The composite microparticles were spherical particles with some deformities (Figure 1).

Based on powder X-ray diffraction tests, the whole product was amorphous, which promotes the rapid dissolution. The X-ray diffractograms of the initially amorphous polymer, the crystalline active ingredient and the composite microparticles (SD1) are shown in Figure 2.

### 3.2. Size of the Oleogels

The HPMC polymer used in the experiments had double tasks. On one hand, as a carrier polymer, it helped stabilize the amorphous form of the active ingredient, and on the other hand, the composite microparticles could be dispersed homogenously into the oleogel. The active ingredient in its crystalline form settled at the bottom of the oleogel and it was not possible to produce a homogeneous product, hence it could not be prepared as a reference sample for the ex vivo and in vivo investigations.

The particle size distribution of composite microparticles in gels can be seen in Table 1. The volume mean particle size of the dried product of SD1 was 2.67 µm, while in LO5 gel and gel containing menthol penetration enhancer (LO5M) it was 2.99 µm and 2.85 µm, respectively. The mean size values and the size distributions indicate that the incorporation of microparticles in gel did not influence the particle size significantly (Figure 3).

### 3.3. Raman Microscopic Investigations

As a first step, comparison of the spectra of the active ingredient and its formulations was performed for determining the evaluation method of the Raman mapping of the skin samples (Figure 4).

Figure 4 shows that the spectral properties of the active ingredient appear to a small extent in the oleogel formulations. The reason for this was that the strong spectral properties of the carrier system masked the spectral properties of the active substance in the given concentration. According to the results, the fingerprint range of the spectra of the formulations (LO5, LO5M) was taken into account for the evaluation of Raman mapping of the skin samples. Using these spectra, a correlation map was created to detect the penetration of the active substance into the skin.

The correlation map on the Raman map of the untreated abdominal skin sample was also created, which resulted in the maximum correlation value of 0.14. This value was chosen as the maximum value of the correlation measure scale for the treated skin samples. Figure 5 shows that the correlation maximum value is higher for the treated samples. The presence of the active substance in certain layers of the skin was indicated by a color scale. From blue to red it indicates the increasing presence of the active substance, which is related to the degree of correlation.

From the evaluation of the Raman maps of the skin samples, it can be concluded that the oleogels containing the active ingredient LC are highly enriched in the epidermis layer of the skin in both cases (Figure 5). The dermis layer was measured only at a small depth, the active ingredient penetrated in this layer to a lesser extent.

By comparing the spectral properties of the active substance and the oleogel formulations, it was established that the evaluation method was applicable to the penetration measurements of the active substance in the skin; therefore, correlation maps were prepared to show the distribution of the active substance in the individual layers of the skin. From the evaluation of the Raman correlation maps, it can be stated that the prepared oleogels acted as good vehicles of the active ingredient LC, which was enriched largely in the epidermis and the upper region of dermis, with and without menthol penetration enhancer. Since urticaria affects this part exactly, these preparations can be suitable for its treatment.

### 3.4. In Vivo Experiments

The anti-inflammatory study was carried out by croton oil-induced edema in mice. Totals of 5 µL and 10 µL of LO10 reduced the edema thickness considerably, while the 20 µL/ear of LO10 oil gel exceeded the positive control effect of 20 µL/ear Fenistil, as depicted in Figure 6. LO10M gel with the same LC concentration as LO10, additionally including menthol, reduced the edema more efficiently, even at a lower dose of oil gel compared to 20 µL/ear Fenistil (Figure 7). The increase in LO10M gel dose did not change the anti-inflammatory effect substantially.

## 4. Conclusions

In this study, levocetirizine dihydrochloride active ingredient was nano spray-dried with HPMC polymer carrier. The microparticles had an average diameter of 2.67 and 2.59 µm. The active ingredient was embedded in an amorphous state in the amorphous polymer according to XRD investigations. The obtained HPMC-encapsulated drug microparticles were mixed into topical gels in order to develop an effective formulation for the treatment of allergic edema. The nano spray-dried powder could be easily and homogeneously dispersed in dermal oil gel composed of Miglyol 812 gelated by Dermofeel viscolid powder. Qualitative ex vivo penetration studies using a Raman microscopic correlation mapping proved the epidermal and also the upper dermal enrichment of drug. Furthermore, the addition of menthol penetration enhancer enriched the drug in the upper dermis of human abdominal skin, which is beneficial considering the expected effect. The spray-dried polymer–drug composite microparticles enabled an easy preparation of a homogeneous dispersion of the drug in oleogel. The prepared oleogels provided an enhanced skin penetration of levocetirizine dihydrochloride. The qualitative skin penetration results were also investigated in vivo. The tests on the mouse model of ear edema induced by croton oil showed similar anti-inflammatory efficiency to commercial Fenistil gel; moreover, the gel volume, and consequently the applied drug amount, could be decreased considerably with the addition of menthol.

## Figures and Tables

**Figure 1 pharmaceutics-14-02577-f001:**
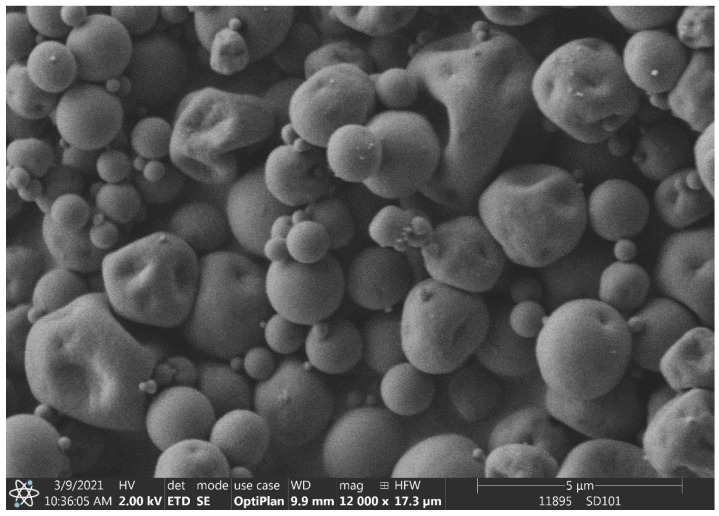
SEM image of SD1 nano spray-dried sample.

**Figure 2 pharmaceutics-14-02577-f002:**
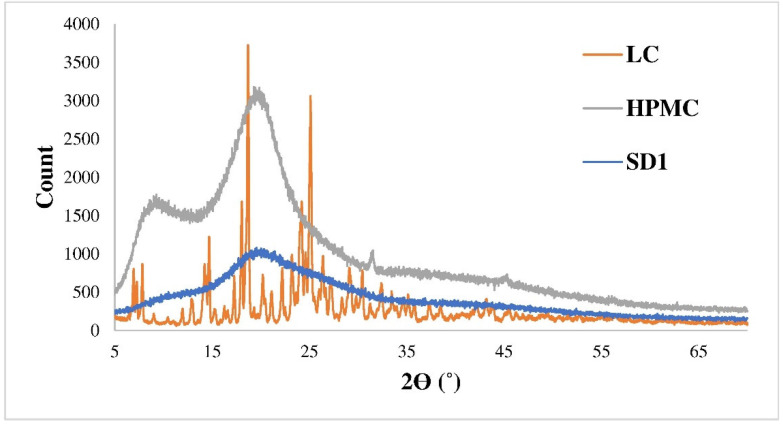
X-ray diffractogram of LC, HPMC, and SD1.

**Figure 3 pharmaceutics-14-02577-f003:**
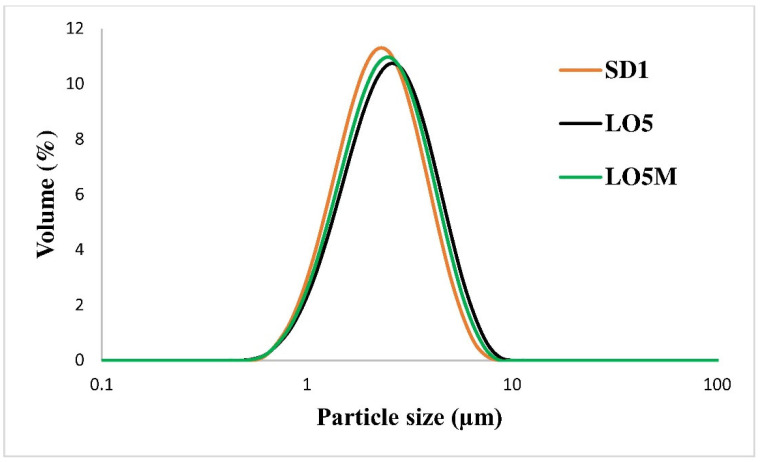
Particle size distribution of composite particles in SD1, LO5, and LO5M samples.

**Figure 4 pharmaceutics-14-02577-f004:**
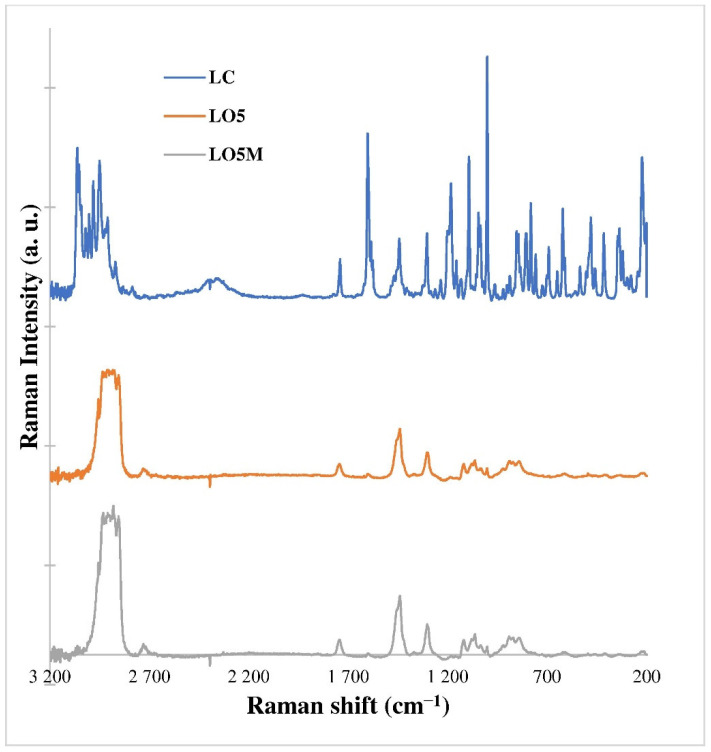
Raman spectra of LC active ingredient and oleogel formulations of LO5 and LO5M.

**Figure 5 pharmaceutics-14-02577-f005:**
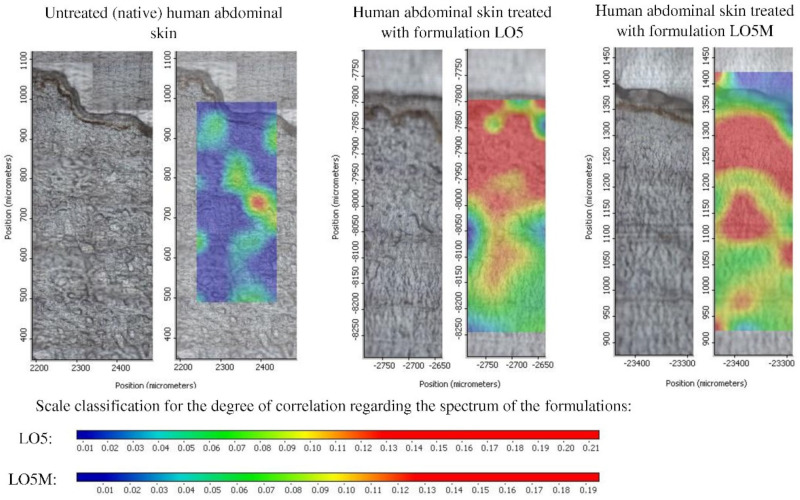
Raman correlation map of human abdominal skin samples treated with drug-loaded oleogel formulations.

**Figure 6 pharmaceutics-14-02577-f006:**
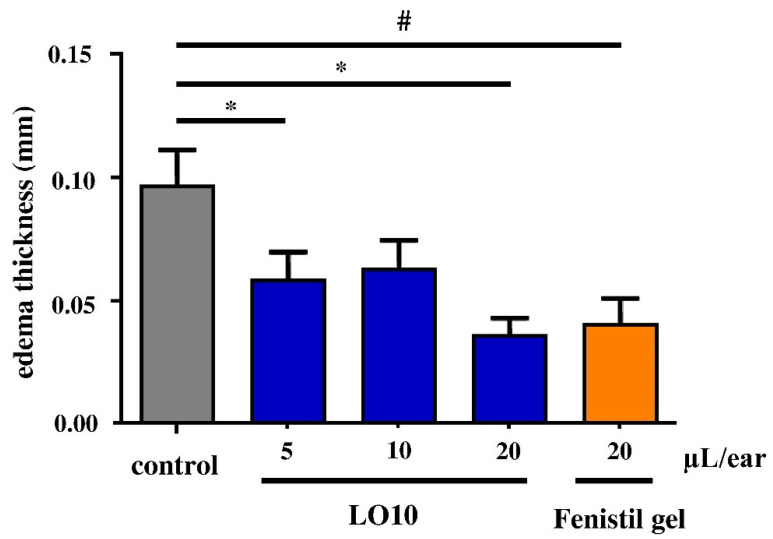
In vivo test of LO10 oleogel. * *p* ≤ 0.05; # *p* ≤ 0.05.

**Figure 7 pharmaceutics-14-02577-f007:**
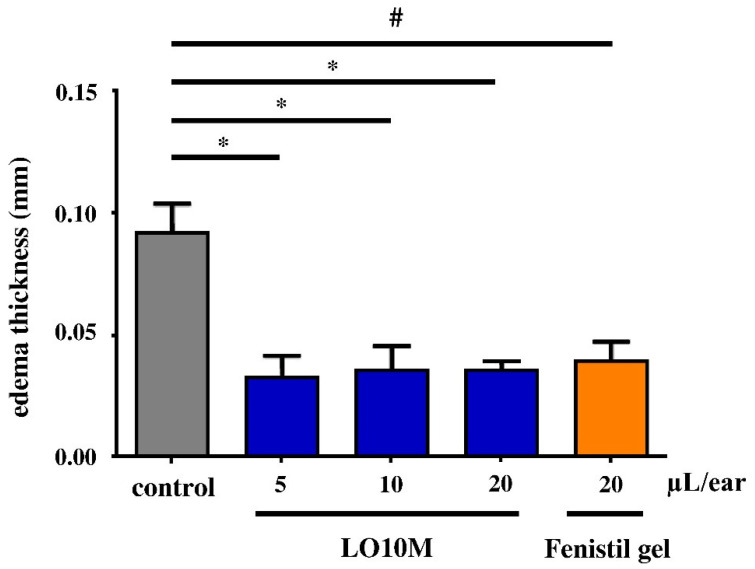
In vivo test of LO10M oleogel. * *p* ≤ 0.05; # *p* ≤ 0.05.

**Table 1 pharmaceutics-14-02577-t001:** Particle size distribution of the LC and the spray-dried products.

Sample	Dv10 (µm)	Dv50 (µm)	Dv90 (µm)	D(4,3) (µm)	Span
LC (sonicated)	1.44	6.07	19.78	10.68	3.02
SD1	1.31	2.43	4.39	2.67	1.27
SD2	1.31	2.39	4.14	2.59	1.18
LO5 *	1.39	2.71	4.99	2.99	1.33
LO5M *	1.35	2.58	4.72	2.85	1.31

* Particle size distribution of microparticles in the oleogel.

## Data Availability

Data are contained within the article.

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
