# Peer review of "Topical Formulation of Nano Spray-Dried Levocetirizine Dihydrochloride against Allergic Edema"

_pharmaceutics, 2022, doi:10.3390/pharmaceutics14122577_

Round 1
Reviewer 1 Report
The authors reported a study where they encapsulated Levocetirizine dihydrochloride as active substance inside HPMC via spray drying technique. The manuscript is well formulated and documented with clear outcome. However, the novelty aspect of this study is very low and no statement was inserted. The polymer the authors used is well known and well -used; the encapsulation method as well. The authors did not defend their results and make a comparison with what was done before.
Author Response
Reviewer: However, the novelty aspect of this study is very low and no statement was inserted. The polymer the authors used is well known and well -used; the encapsulation method as well. The authors did not defend their results and make a comparison with what was done before.
Authors’ reply: We thank the reviewer for the helpful comment. We have completed the 1. Introduction section with further details and references. Results and Conclusion sections have been also completed accordingly. Moreover, after the literature survey, we emphasised more clearly the novelty of our study as follows:
“It can be established based on the literature data dealing with cetirizine topical application that in previous studies the LC was in dissolved form in different type of hydrogel or oleogels with substantial water content. In those cases, the formulation requires several excipients as surfactant, preservative, co-solvent, penetration enhancer to reach the expected skin penetration. However, in our study we developed oleogels without adding water and/or surfactants. The commercially available products of cetirizine are solid, drop or syrup formulas. For example, the ZYRTEC® Liquid Gels is a capsule of 10 mg/capsule content for oral use [21]. Our aim was to prepare a new gel formulation of levocetirizine dihydrochloride for topical administration against allergic edema like urticaria. Oleogel was chosen as base for the product as promising vehicle for transdermal delivery. Composite microparticles with HPMC polymer were prepared by nano spray-dryer for ensuring the easy incorporation of the solid state drug in oleogel. Gels were prepared without and with menthol as penetration enhancer for dermal administration. Menthol is a common excipient in different cosmetics and topical drug formulations. It has a dual effect: acts as a local anaesthetic and a penetration enhancer [22, 23]. The ex vivo cutaneous penetration of drugs was investigated on human skin samples by confocal Raman microscopic method and in vivo tests were done on the mouse model of croton oil-induced ear edema. The efficacy was compared to commercially available Fenistil gel containing dimetindene maleate as active ingredient.”
Reviewer 2 Report
Dear Authors,
In this work, the authors worked on the preparation of nano spray-dried HPMC polymeric microparticles containing levocetirizine dihydrochloride active ingredient for topical administration against allergic edema disease. Morphological and XRD characterization of polymeric microparticles, in vitro confocal Raman microscopic investigations, and in vivo tests for the resultant materials were performed to describe features of the topical formulation of the nano-spray-dried levocetirizine dihydrochloride against allergic edema. The work is interesting, both the experiments and the results achieved are carefully organized and references are well-listed. The results earn probably wide interest justifying publication in Pharmaceutics. Therefore, I suggest publication as subject to the following minor remarks;
In line 15, the authors should add the abbreviation “HPMC” for “Hydroxyl propyl methyl cellulose”.
In line 22, the authors should use the “confocal Raman” phrase.
In line 73, Is the terminology used “ex vivo” or “in vitro”, the authors should make sure to use the right terminology.
In line 82, Did the authors use water during the experiments?
Between lines 119 and 122, the Materials used in this part of the experimental work should be mentioned in the “2.1 Materials” section.
In line, the authors should be careful about terminology. Is this investigation “In vitro” or “ex vivo”, please make this part clear for better understanding.
In line 153, Is this Raman microscopic investigation performed in In vitro or Ex vivo conditions? The authors should make this part clear for better understanding. The authors check the “In vitro Raman” description in the abstract.
In line 214, The authors should calculate PDI values for the samples examined and put the calculation results into Table 1 to make clear for better understanding of the method’s capability to make microparticles.
In line 216, a better representative SEM image should be used in this manuscript to show spherical micro-scaled particles with particle distribution.
In line 288, the authors should describe the core-shell layers of microparticles obtained. Is the topical gel encapsulated into the HPMC polymeric shell structure or active ingredient encapsulated HPMC particles mixed into the topical gel?
Between lines 287 and 299, the authors should present what conclusions were achieved depending on structural, XRD, and ex vivo investigations in the “4. Conclusions” part.
Regards,
Author Response
Reviewer: In line 15, the authors should add the abbreviation “HPMC” for “Hydroxyl propyl methyl cellulose”.
Authors’ reply: It has been completed as follows:
“Levocetirizine dihydrochloride active ingredient was microencapsulated using nano spray-drying technology for preparing microparticles containing topical gel against edema. Hydroxyl propyl methyl cellulose (HPMC) was used as a carrier polymer during spray-drying.”
Reviewer: In line 22, the authors should use the “confocal Raman” phrase.
Authors’ reply: It has been done. “Qualitative ex vivo penetration studies using a confocal Raman microscopic correlation map-ping was executed on human abdominal skin.”
Reviewer: In line 73, Is the terminology used “ex vivo” or “in vitro”, the authors should make sure to use the right terminology.
Authors’ reply: The penetration tests are considered ex vivo as human skin samples were used. The sentence has been corrected:
In line 94:
„The ex vivo cutaneous penetration of drugs was investigated on human skin samples by confocal Raman microscopic method and in vivo tests were done on the mouse model of croton oil-induced ear edema.”
Reviewer: In line 82, Did the authors use water during the experiments?
Authors’ reply: It has been completed in 2.2. Materials section, lines 120-123:
“Solutions for the experiments were prepared as follows: 1 % (w/w) of LC and 1 % (w/w) of HPMC in MilliQ water were prepared by magnetic stirring and used without filtering.”
Reviewer: Between lines 119 and 122, the Materials used in this part of the experimental work should be mentioned in the “2.1 Materials” section.
Authors’ reply: It has been done.
Reviewer: In line, the authors should be careful about terminology. Is this investigation “In vitro” or “ex vivo”, please make this part clear for better understanding.
Authors’ reply: Section 2.4. has been revised accordingly.
Reviewer: In line 153, Is this Raman microscopic investigation performed in In vitro or Ex vivo conditions? The authors should make this part clear for better understanding. The authors check the “In vitro Raman” description in the abstract.
Authors’ reply: Section 2.5. has been revised,
Reviewer: In line 214, The authors should calculate PDI values for the samples examined and put the calculation results into Table 1 to make clear for better understanding of the method’s capability to make microparticles.
Authors’ reply: The polydispersity index is calculated by the DLS (dynamic ligtht scattering) software by fitting a polynomial function to the logarithm of the G1 correlation function of the measured values from the coefficient of the linear (b) and the quadratic term (c) based on the following relationship: PdI = 2c/b2. Although for laser diffraction particle size analysis this information is not available. One of the common values used for laser diffraction results is the span, with the following definition: span: (Dv90-Dv10) /Dv50. the span data have been added into Table 1, which is used to calculate the size distribution width.
Reviewer: In line 216, a better representative SEM image should be used in this manuscript to show spherical micro-scaled particles with particle distribution.
Authors’ reply: Unfortunately, we do not have a more representative image; nevertheless, in our opinion, the SEM image (Figure 1) in the article is in accordance with the size distribution provided by laser diffraction (Figure 3, SD1). In the SEM image both smaller and larger particles are shown, which are consistent with the Dv10, Dv50 and Dv90 values (Table 1).
Reviewer: In line 288, the authors should describe the core-shell layers of microparticles obtained. Is the topical gel encapsulated into the HPMC polymeric shell structure or active ingredient encapsulated HPMC particles mixed into the topical gel?
Authors’ reply: In the cited paper [19] at page 77. the following was written about the particles obtained by spray drying:” A microcapsule can be either an individually coated solid particle or liquid droplet, or a matrix containing many small, fine core particles. Matrix microcapsules containing drug substance and a biodegradable polymer are usually prepared by spray drying in order to obtain controlled drug release formulations.” The microparticles of our study probably have a matrix structure.
Reviewer: Between lines 287 and 299, the authors should present what conclusions were achieved depending on structural, XRD, and ex vivo investigations in the “4. Conclusions” part.
Authors’ reply: The Section 4. Conclusion part has been corrected as follows:
In this study, levocetirizine dihydrochloride active ingredient was nano spray-dried with HPMC polymer carrier. The obtained, spherical like microparticles had an average diameter of 2.67 and 2.59 µm. The active ingredient was embedded in amorphous state in the amorphous polymer according to XRD investigations. The obtained HPMC encapsulated drug microparticles were mixed into topical gels in order to develop an effective formula-tion for the treatment of allergic edema. The nano spray-dried composite powder could be easily and homogeneously dispersed in dermal oil gel composed of Miglyol 812 gelated by Dermofeel viscolid powder. Qualitative ex vivo penetration studies using a Raman mi-croscopic correlation mapping proved the epidermal and also the upper dermal enrich-ment of drug. Furthermore, the addition of menthol penetration enhancer enriched the drug in the upper dermis of human abdominal skin, which is beneficial considering the expected effect. The spray-dried polymer-drug composite microparticles allowed an easy preparation of a homogeneous dispersion of the drug in oleogel. The prepared oleogels provided an enhanced skin penetration of levocetirizine dihydrochloride. The qualitative skin penetration results were further investigated in vivo. The tests on the mouse model of ear edema induced by croton oil showed similar anti-inflammatory efficiency to commercial Fenistil gel; moreover, the gel volume, consequently the applied drug amount could be decreased considerably with the addition of menthol.
Round 2
Reviewer 1 Report
The authors answered to the addressed queries and updated the manuscript accordingly.